# Simulation Analysis on Electromagnetic Vibration and Noise of Novel Mechatronic-Electro-Hydraulic Coupler

**Baoquan Liu** [1,2]**, Tiezhu Zhang** [1,2]**, Hongxin Zhang** [1,2,*]**, Zhen Zhang** [1,2] **and Yang Cao** [1,2]

1    College of Mechanical and Electrical Engineering, Qingdao University, Qingdao 266071, China
2    Power Integration and Energy Storage Systems Engineering Technology Center (Qingdao),
     Qingdao 266071, China
*    Correspondence: zhx@qdu.edu.cn; Tel.: +86-135-7386-5229

**Abstract:** The mechatronic-electro-hydraulic coupler (MEHC) is a novel type of multisource coupling power device which integrates a traditional permanent magnet synchronous motor with a swash plate axial piston pump/motor to realize the mutual conversion of electrical energy, mechanical energy, and hydraulic energy. In order to improve the MEHC's noise, vibration, and harshness performance, an electromagnetic vibration and noise simulation analysis was performed with a six-pole 36-slot motor as the research object. Firstly, the spatial order and frequency of the radial electromagnetic force were deduced by an analytical method. Subsequently, the electromagnetic field was simulated, and the electromagnetic force was extracted via a fast Fourier transform using Ansys/Maxwell software for the numerical verification. Thereafter, the harmonic response module coupled the electromagnetic field with the structural field for the harmonic response analysis. Ultimately, the research results were imported into the harmonic acoustics module for a noise simulation analysis and two different-shape magnetic isolation bridges optimization schemes were proposed. The results suggested that both optimisation solutions could effectively reduce motor vibration and noise. Scheme one reduced the maximum noise by about 6.5% and scheme two by 10.4%. The analysis process and conclusion provide a theoretical basis for the vibration and noise analysis of permanent magnet synchronous motors with different integer slots.

**Keywords:** mechatronic-electro-hydraulic coupler; electromagnetic force; vibration and noise; magnetic isolation bridges optimisation

## 1. Introduction

### 1.1. Research Motivation

Swash plate axial piston pumps/motors are widely used in industry and have the advantages of a high efficiency, compactness, and reliability [1]. As a power source for electric vehicle drive systems, interior permanent-magnet synchronous motors (IPMSM) offer a high power density, wide speed range, high efficiency and other good properties [2]. The mechatronic-electro-hydraulic coupler (MEHC) integrates a permanent magnet synchronous motor and a swash plate axial piston pump/motor to realise the mutual conversion of electrical energy, mechanical energy, and hydraulic energy. The noise from the MEHC is mainly generated by swash plate axial piston pumps/motors [3] and noise from permanent magnet synchronous motors [4], where the noise generated by the axial piston pump can be divided into structural noise, fluid noise, and airborne noise [5], and the noise generated by the permanent magnet synchronous motor is mainly divided into electromagnetic noise, mechanical noise, and aerodynamic noise [6]. Electromagnetic noise is mainly generated by the radial electromagnetic force acting between the stator and rotor of the motor and is the main part of the motor vibration and noise [7], which is also the main content of the analysis in this paper. The MEHC as a vehicle drive device, the motor vibration, and the noise will affect the ride comfort and operational reliability of new



energy vehicles, so reducing motor vibration noise is essential to improve the comfort and operational stability of new energy vehicles.

### 1.2. Literature Review

As people demand more and more comfortable car rides, NVH (noise, vibration, and harshness) performance indicators are becoming increasingly important in the automotive industry [8,9]. Compared to conventional vehicles, the electric vehicle (EV) motor drive system replaces the internal combustion engine as the primary source of vibration and noise [10]. The radial electromagnetic force, generated by the air gap's magnetic field, is a rotational force wave that varies with time and space. Its component in the radial direction of the motor acts on the motor stator teeth, causing the stator core to vibrate in response. Several scholars have studied the electromagnetic force of motors. Ref. [11] analysed the radial and axial electromagnetic force of disc motors at low temperatures, and the results proved that when the motor rotor was radially eccentric, the radial electromagnetic force tried to pull the rotor back to the centre position, and the radial electromagnetic force increased with the increase of speed and radial eccentric displacement. The authors in [12] proposed a fast and accurate method for calculating the radial electromagnetic force density of surface-mounted permanent-magnet synchronous motors and used an auxiliary slot in the stator to weaken the radial electromagnetic force wave. In [13], an analytical model considering the modulation effect of the stator teeth on the radial electromagnetic force in a permanent-magnet synchronous motor was developed, which could study the radial vibration characteristics of the stator of a permanent magnet synchronous motor promptly and could also effectively reduce the stator vibration amplitude. The authors in [14] studied the radial electromagnetic force and vibration distribution of an external rotor's switched reluctance motor for electric bicycles and analysed the effect of the switching call on rotor vibration. The analysis found that adjusting the opening angle could reduce the high rate of change of the radial electromagnetic force waveform at the pole angle. The authors in [15] investigated the effect of different deformations on the electromagnetic force wave amplitude and radial vibration characteristics of built-in permanent-magnet synchronous motors. The analysis compared stator elliptical deformation, rotor centrifugal deformation, and the effect of both deformations on the air gap's magnetic field and radial and tangential electromagnetic forces, and the simulation results showed that air-gap deformation had a significant impact on vibration.

In order to suppress motor vibration and noise, [16] installed a stator bridge on the surface of the fractional slot of a permanent magnet synchronous motor to reduce vibration and noise. They proposed a control structure of a fixed stator bridge that could effectively reduce the vibration and noise of a permanent-magnet synchronous motor. The authors in [17] analysed the primary sources and characteristic parameters of the zero-order electromagnetic force of the motor, proposed a stator and rotor structure optimisation method to increase the intrinsic frequency of the motor and reduce the harmonic content of the electromagnetic force, and verified through experiments that the method could effectively reduce the vibration and noise of the motor. The authors in [18] studied several fractional-slot permanent-magnet synchronous motor topologies in order to reduce the vibration noise effectively and verified through experiments the feasibility of the studied solutions. The authors in [19] predicted the vibration noise level of a powertrain under the simultaneous action of gear meshing forces and electromagnetic forces, and the experimental values were in general agreement with the predicted results and pointed out that gear meshing forces and electromagnetic forces had a significant influence on the noise level of the structurally integrated powertrain. The authors in [20] analysed the mechanism of motor vibration noise generation and reduced motor vibration noise by reducing the electromagnetic force amplitude at primary frequencies to an optimal motor structure. The authors in [21] proposed two methods of optimising electromagnetic forces to reduce vibration noise. The first was to optimise the radial electromagnetic force amplitude by optimising the slot width and the radius of the permanent magnet's fillet. The second was optimising the phase

distribution of the radial electromagnetic force along the axial direction through the rotor's continuous and segmented sloping poles. Through a comparison, it was found that both of them had different effects on suppressing motor noise. In [22], an analytical model of the radial electromagnetic force of a permanent-magnet synchronous motor considering the segmented sloping poles of the rotor was developed, and the influence of the segmented sloping poles on the electromagnetic vibration noise of the motor was analysed and summarised. In [23], a finite element method was used to investigate the vibration noise of a fractional-slot permanent-magnet synchronous motor. Via vibration noise suppression through an inclined rotor pole, stator slot wedge, and stator auxiliary slot, the comparison results found that the addition of a stator slot wedge had the best suppression effect. In addition, the stator tooth tip arc offset structure [24], different stator tooth chamfers [25], the use of rotor inclined poles [26,27], and changes to the motor rotor slot structure [28] can all be effective at reducing motor vibration noise.

### 1.3. Challenges and Problems

Many challenges and issues remain with current MEHCs' electromagnetic vibration and noise analysis. First, most of the current multisource power coupling devices are in the research design and simulation stage, and only a few research teams have realized prototype experiments. Therefore, this paper is essential for the design aspects of the NVH study of the MEHC.

Secondly, current simulation software has certain delays and errors, so the simulation results do not fully reflect the actual working conditions.

Finally, as the vehicle's permanent-magnet synchronous motor requires a strong overload capacity and wide speed range, as well as a high power density and low weight, the vibration noise problem of the permanent-magnet synchronous motor has begun to surface, as high-frequency electromagnetic whine seriously affects the car ride comfort. Therefore, it is of great significance to analyse and suppress the motor's vibration noise.

### 1.4. Contributions of this Work

This paper focuses on the following work and contributions in response to these issues:

(1) A novel new electro-mechanical-hydraulic coupling power unit is proposed, which integrates a conventional permanent-magnet synchronous motor and a swashplate axial piston pump/motor to achieve arbitrary conversion between mechanical energy, electrical energy, and hydraulic energy.

(2) The possible spatial order and temporal frequency of the electromagnetic forces are predicted using an analytical method and verified by electromagnetic simulations.

(3) An analysis of the modal inherent frequency of the motor stator at each order is performed. The MEHC's vibration and noise are analysed, and two optimisation schemes are proposed to suppress the vibration and noise. It is verified through simulation that both optimisation schemes can effectively suppress the motor vibration and noise.

### 1.5. Organisation of the Paper

The rest of the paper is organised as follows: Section 2 describes the structure and energy conversion process of the MEHC. Section 3 predicts the radial electromagnetic force's spatial order and temporal frequency using an analytical method and verifies them by simulation with electromagnetic simulation software. Section 4 analyses the inherent modal frequency of each order of the motor stator. Section 5 analyses the motor vibration and noise and proposes two optimisation solutions to suppress vibration and noise.

## 2. Structure and Working Principle of the MEHC

The structure of the MEHC consists of mechanical components, swashplate hydraulic components, motor components, and variable mechanisms, which have the advantages of a compact structure, high efficiency, strong adaptation to the working environment, etc. The overall structure is shown in Figure 1a.

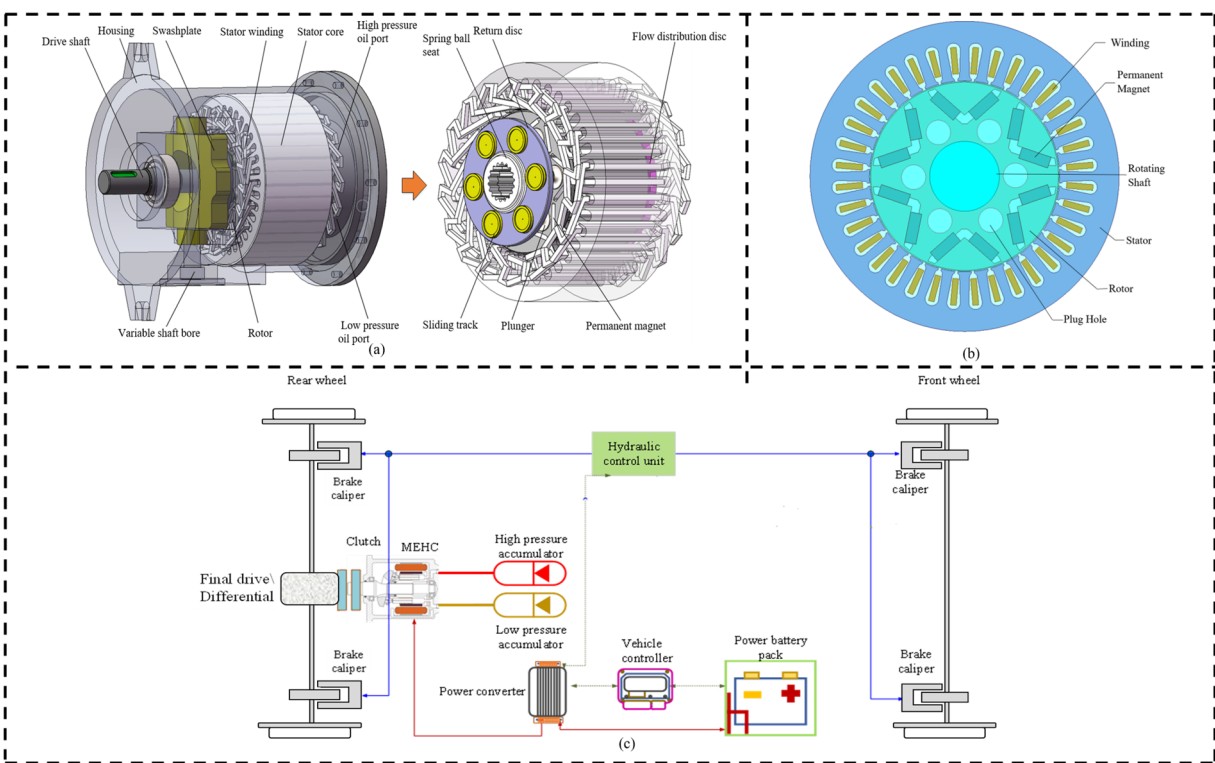

**Figure 1.** MEHC's application schematic: (**a**) MEHC's structure; (**b**) MEHC's two-dimensional model of a motor; (**c**) MEHHC's application schematic in vehicle.

The multiple energy conversion of the MEHC allows it to meet the requirements of a wide range of vehicle operating conditions. The principle of the MEHC's coupling in vehicle applications is shown in Figure 1c. When mechanical energy is converted into hydraulic energy, the mechanical energy acts on the driveshaft and drives the piston pump's rigid body into rotational motion. In this case, the plunger follows the rotational motion of the rigid body while reciprocating in a linear motion with the cooperation of the sliding track, swashplate, and other components. During the suction process, the plunger moves outwards, and the volume of the plunger chamber increases until it enters the outlet chamber area. Under the oil pressure, the plunger moves inwards, the volume of the plunger chamber decreases, the pressure in the chamber increases and high-pressure oil is output. When the hydraulic energy is converted into mechanical energy, the high-pressure oil enters the plunger cavity, and the pressure inside the cavity increases so that the plunger moves outwards, driving the rigid body to rotate with the cooperation of the swashplate, which drives the driveshaft to rotate and output mechanical energy outwards. Under both conditions, the rigid body completes one energy conversion cycle for each rotation, enabling the interconversion of mechanical and hydraulic energy. According to the requirements of the working conditions, the angle of inclination of the swashplate can be adjusted through the variable mechanism to achieve an increase or decrease in the output of mechanical or hydraulic energy to meet the requirements of the working conditions.

When mechanical energy is converted into electrical energy, the mechanical energy drives the motor rotor (piston pump rigid body) through the drive shaft. The permanent magnet rotates to cut the three-phase winding to produce alternating electric potential, which outputs current through the wire to the outside. When electrical energy is converted into mechanical energy, a three-phase alternating current is input through the wire, and the interaction between the three-phase winding in the stator slot and the stator core generates a rotating magnetic field, which drives the rotor to drive the driveshaft to rotate and output mechanical energy to the outside. When the MEHC is working in single-to-multiple or

multi-to-single energy conversion, the operating mode can be selected according to the actual working environment.

## 3. Parametric Analysis of Electromagnetic Force Wave Characteristics of MECH

### 3.1. Theoretical Analysis of the Characteristic Parameters of Electromagnetic Force Waves

The MEHC's motor is a built-in permanent-magnet synchronous motor. Without considering the saturation of the core reluctance, the motor's air-gap flux density expression is:

$$b(\theta, t) = f(\theta, t)\Lambda(\theta) \tag{1}$$

where $\theta$ is the spatial Angle, $t$ is time, $b(\theta,t)$ is the air-gap magnetic density of the motor, $f(\theta,t)$ is the stator–rotor synthetic magnetic potential, and $\Lambda(\theta)$ is the relative magnetic permeability function. When a sinusoidal current is supplied, the magnetomotive force $f(\theta,t)$ is:

$$f(\theta, t) = f_\mu(\theta, t) + f_v(\theta, t) = \sum F_\mu \cos(\mu p\theta - \mu\omega t - \varphi_\mu) + \sum F_v \cos(vp\theta - \omega t - \varphi_v) \tag{2}$$

where $f_\mu(\theta,t)$ is the rotor's permanent magnets' magnetodynamic potential and the rotor's spatial harmonic number is $\mu = 1,3,5...$, $f_v(\theta,t)$ is the stator's magnetomotive force with spatial harmonic number $v = 1,5,7...$, $F_\mu$ is the rotor $\mu$th harmonic potential amplitude, $F_v$ is the stator $v$th harmonic potential amplitude, $\omega$ is the angular velocity corresponding to the fundamental frequency, $\varphi_\mu$ is the rotor's magnetic potential phase, $\varphi_v$ is the stator's magnetic potential phase, and $p$ is the number of motor pole pairs. When the rotor surface is smooth, only the inner side of the stator is slotted, and the relative permeability function $\Lambda(\theta)$ can be expressed as:

$$\Lambda(\theta) = \Lambda_0 + \sum_{k=1}^{\infty} \Lambda_k \cos(kZ\theta) \tag{3}$$

where $Z$ is the number of stator slots, $k = 1,2,3...$

The effect of the radial electromagnetic force on the motor vibration noise is much greater than that of the tangential electromagnetic force, so the tangential electromagnetic force can be ignored. The expression for the radial electromagnetic force is:

$$b_r(\theta, t) \approx b(\theta, t) \tag{4}$$

Combining Equations (1)–(4) yields:

$$
\begin{aligned}
b_r(\theta, t) &= \left[\sum F_\mu \cos(\mu p\theta - \mu\omega t - \varphi_\mu) + \sum F_v \cos(vp\theta - \omega t - \varphi_v)\right] \cdot \left[\Lambda_0 + \sum_{k=1}^{\infty} \Lambda_k \cos(kZ\theta)\right] \\
&= \Lambda_0 \sum F_\mu \cos(\mu p\theta - \mu\omega t - \varphi_\mu) + \sum\sum \Lambda_k \cos(kZ\theta)F_\mu \cos(\mu p\theta - \mu\omega t - \varphi_\mu) \\
&\quad + \Lambda_0 \sum F_v \cos(vp\theta - \omega t - \varphi_v) + \sum\sum \Lambda_k \cos(kZ\theta)F_v \cos(vp\theta - \omega t - \varphi_v) \\
&= \Lambda_0 \sum F_\mu \cos(\mu p\theta - \mu\omega t - \varphi_\mu) + \sum\sum \frac{\Lambda_0 F_\mu}{2} \cos[(\mu p \pm kZ)\theta - \mu\omega t - \varphi_\mu] \\
&\quad + \Lambda_0 \sum F_v \cos(vp\theta - \omega t - \varphi_v) + \sum\sum \frac{\Lambda_0 F_v}{2} \cos[(vp \pm kZ)\theta - \omega t - \varphi_v]
\end{aligned}
\tag{5}
$$

The above equation can be simplified as follows:

$$b_r(\theta, t) = (b_{p1} + b_{p2}) + (b_{s1} + b_{s2}) \tag{6}$$

where $b_{p1}$ is the rotor's magnetic field produced by constant permeability modulation, $b_{p2}$ is the rotor's magnetic field produced by slotted permeability modulation, $b_{s1}$ is the stator's magnetic field produced by constant permeability modulation, and $b_{s2}$ is the stator's magnetic field produced by slotted permeability modulation.

According to Maxwell's tensor method, the expression for the radial electromagnetic force per unit area is:

$$
\begin{aligned}
F_r(\theta, t) &= \frac{b_r(\theta,t)^2}{2\mu_0} = \frac{\left[(b_{p1}+b_{p2})+(b_{s1}+b_{s2})\right]^2}{2\mu_0} \\
&= \frac{1}{2\mu_0}\left(b_{p1}^2 + b_{p2}^2 + 2b_{p1}b_{p2} + b_{s1}^2 + b_{s2}^2 + 2b_{s1}b_{s2} + 2b_{p1}b_{s1} + 2b_{p1}b_{s2} + 2b_{p2}b_{s1} + 2b_{p2}b_{s2}\right) \\
&= \frac{1}{2\mu_0}\left(b_{p1}^2 + b_{p2}^2 + 2b_{p1}b_{p2}\right) + \frac{1}{2\mu_0}\left(b_{s1}^2 + b_{s2}^2 + 2b_{s1}b_{s2}\right) + \frac{1}{\mu_0}\left(b_{p1}b_{s1} + b_{p1}b_{s2} + b_{p2}b_{s1} + b_{p2}b_{s2}\right) \\
&= F_{rp} + F_{rs} + F_{rps}
\end{aligned}
\tag{7}
$$

where $F_{rp}$ is the radial electromagnetic force generated by the rotor's magnetic field, $F_{rs}$ is the radial electromagnetic force generated by the stator's magnetic field, $F_{rps}$ is the radial electromagnetic force generated by the interaction of the stator and rotor's magnetic fields. The radial electromagnetic forces generated by the action of the rotor's magnetic field is:

$$
F_{rp} = \frac{1}{2\mu_0}\left(b_{p1}^2 + b_{p2}^2 + 2b_{p1}b_{p2}\right)
\tag{8}
$$

Substituting Equations (5) and (6), the spatial order and frequency distribution of the radial electromagnetic force generated by the action of the rotor's magnetic field were analysed and obtained, as shown in Table 1.

**Table 1.** Spatial order and frequency distribution of radial electromagnetic forces generated by the action of the rotor's magnetic field.

| | **Air-Gap Magnetic Density Harmonic Count** | **Space Order** | **Frequency** |
|---|---|---|---|
| $b_{p1}\cdot b_{p1}$ | $\mu_1 \neq \mu_2$ | $(\mu_1 \pm \mu_2)p$ | $(\mu_1 \pm \mu_2)f$ |
| | $\mu_1 = \mu_2 = \mu$ | $2\mu p$ <br> $0$ | $2\mu f$ <br> $0$ |
| $b_{p1}\cdot b_{p2}$ | $\mu_1 \neq \mu_2$ | $\mu_1 p \pm \mu_2 p \pm kZ$ | $(\mu_1 \pm \mu_2)f$ |
| | $\mu_1 = \mu_2 = \mu$ | $2\mu p \pm kZ$ <br> $\pm kZ$ | $2\mu f$ <br> $0$ |
| $b_{p2}\cdot b_{p2}$ | $\mu_1 \neq \mu_2$ | $(\mu_1 \pm \mu_2)p$ <br> $\mu_1 p \pm \mu_2 p \pm 2kZ$ | $(\mu_1 - \mu_2)f$ <br> $(\mu_1 + \mu_2)f$ |
| | $\mu_1 = \mu_2 = \mu$ | $2(\mu p + kZ)$ <br> $0$ | $2\mu f$ <br> $0$ |

The radial electromagnetic force resulting from the interaction of the stator and rotor's magnetic fields is:

$$
F_{rps} = \frac{1}{\mu_0}\left(b_{p1}b_{s1} + b_{p1}b_{s2} + b_{p2}b_{s1} + b_{p2}b_{s2}\right)
\tag{9}
$$

Substituting Equations (5) and (6), the spatial order and frequency distribution of the radial electromagnetic force generated by the interaction of the stator and rotor's magnetic fields were analysed, as shown in Table 2.

From Tables 1 and 2, it can be seen that the spatial order of the radial electromagnetic force wave is of order 0 and order 2p, i.e., order 6, order 12, order 18, etc. The force wave frequency of the radial electromagnetic force is an integer multiple of 0 and the electromagnetic frequency 2f, i.e., 2f, 4f, 6f, etc.

**Table 2.** Spatial order and frequency distribution of radial electromagnetic forces resulting from the interaction of stator and rotor's magnetic fields.

| | **Air-Gap Magnetic Density Harmonic Count** | **Space Order** | **Frequency** |
|---|---|---|---|
| $b_{p1} \cdot b_{s1}$ | $M \neq v$ | $(\mu \pm v)p$ | $(\mu \pm 1)f$ |
| | $\mu = v$ | $2\mu p$ <br> $0$ | $(\mu + 1)f$ <br> $(\mu - 1)f$ |
| $b_{p1} \cdot b_{s2}$ | $M \neq v$ | $\mu p \pm vp \pm kZ$ | $(\mu \pm 1)f$ |
| | $\mu = v$ | $2\mu p \pm kZ$ <br> $\pm kZ$ | $(\mu + 1)f$ <br> $(\mu - 1)f$ |
| $b_{p2} \cdot b_{s1}$ | $M \neq v$ | $\mu p \pm vp \pm kZ$ | $(\mu \pm 1)f$ |
| | $\mu = v$ | $2\mu p + kZ$ <br> $+kZ$ | $(\mu + 1)f$ <br> $(\mu - 1)f$ |
| $b_{p2} \cdot b_{s2}$ | $M \neq v$ | $\mu p \pm vp \pm kZ$ | $(\mu \pm 1)f$ |
| | $\mu = v$ | $2\mu p + kZ$ <br> $+kZ$ | $(\mu + 1)f$ <br> $0$ |

### 3.2. Finite Element Simulation of Electromagnetic Forces

The permanent-magnet synchronous motor used in the MEHC was a 6-pole, 36-slot, the three-phase motor had the specific parameters shown in Table 3 and the two-dimensional model of the motor was as shown in Figure 1b.

**Table 3.** Motor parameters.

| Parameter | Specific Values |
|---|---|
| Number of slots/pole pairs | 36/3 |
| Rated power/kW | 18 |
| Rated current/A | 244 |
| Rated speed/(r/min) | 3000 |
| Plunger diameter d/mm | 17 |
| Diameter of plunger distribution circle D/mm | 68 |

Under rated conditions, the change in radial air-gap density with a space angle at 0.01 s when the motor was loaded is shown in Figure 2a, and the change in radial air-gap density at a point on the stator tooth end with time is shown in Figure 2b. Using Maxwell's tensor equation, the variation of the radial electromagnetic force with a space angle at the moment of 0.01 s when the motor was loaded is shown in Figure 3a, and the variation of the radial electromagnetic force at a point on the stator tooth end with time is shown in Figure 3b.

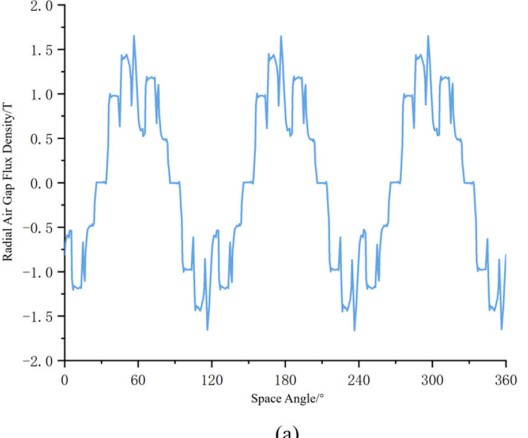
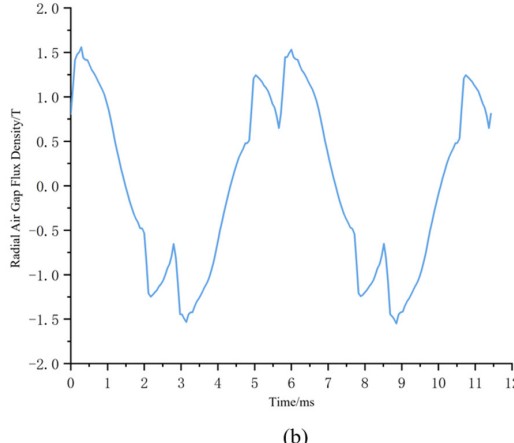

(a)                     (b)

**Figure 2.** Air−gap flux density change: (**a**) 0.01 s changes with space angle; (**b**) change in time at a point on the stator tooth end.

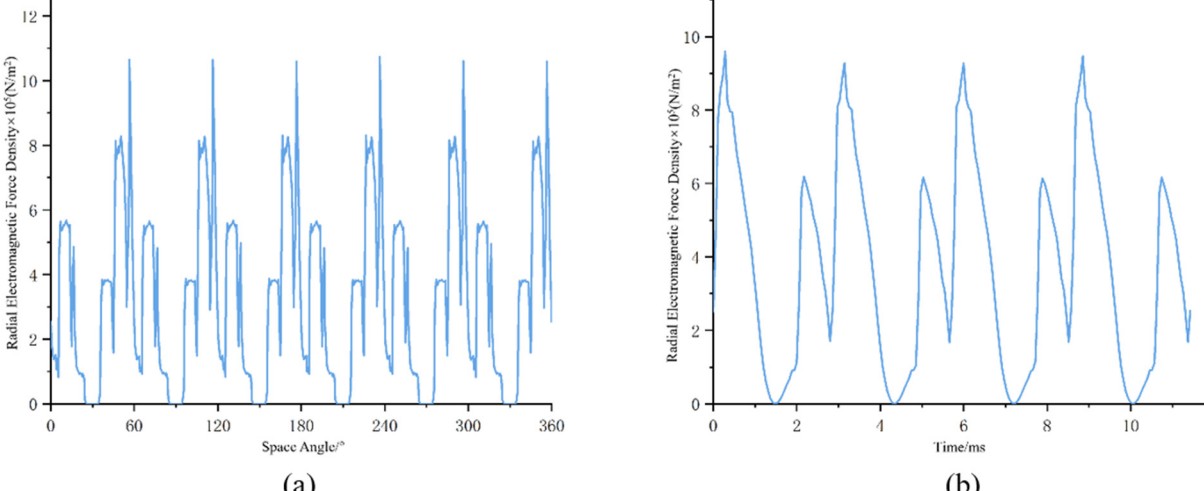

**Figure 3.** Radial electromagnetic force variation: (**a**) 0.01 s changes with space angle; (**b**) change in time at a point on the stator tooth end.

The FFT (fast Fourier transform) decomposition of the radial electromagnetic force density with a space at 0.01 s and the radial electromagnetic force density with time at a point on the stator tooth end is shown in Figure 4a,b. The spatial order of the radial electromagnetic force is mainly distributed at order 0, order 6, order 12, etc., and the variation in time–frequency is mainly distributed at 0, 350 Hz, 700 Hz, 1050 Hz, etc., which is consistent with the results obtained by the analytical method.

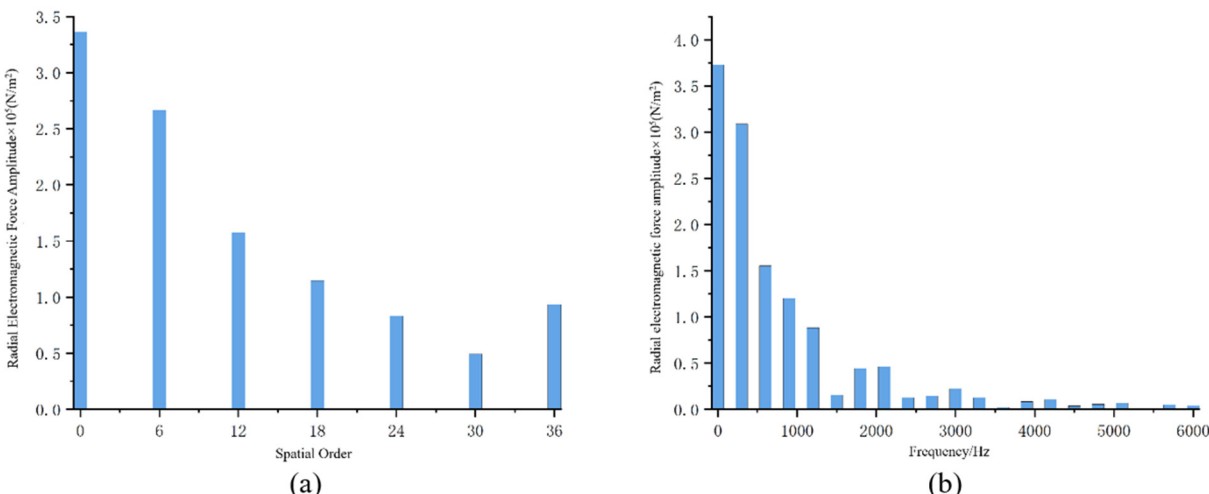

**Figure 4.** (**a**) The 0.01 s radial electromagnetic force distribution with spatial order; (**b**) radial electromagnetic force distribution in time–frequency at a point on the stator tooth end.

## 4. Stator Modal Analysis

Mechanical vibration is generally the common effect of multiple excitation sources superimposed on each other. Each vibration pattern has a vibration frequency, i.e., the inherent frequency. Resonance occurs when an external excitation source causes a structural vibration pattern, and the excitation frequency is close to the intrinsic frequency of the vibration pattern. It is therefore essential to carry out a modal analysis of the stator. The stator structure of the MEHC is shown in Figure 5a, the material used was silicon steel sheet DW310-35, and the modal and intrinsic frequencies of each order are shown in Figure 5b–g.

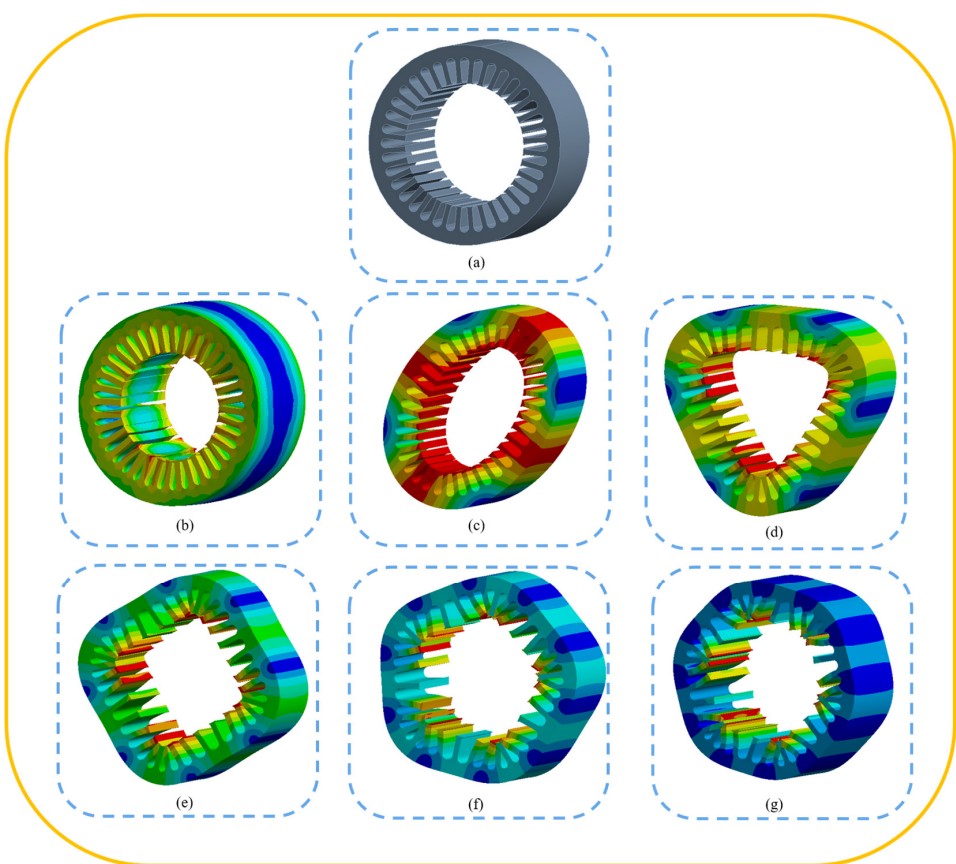

**Figure 5.** (**a**) Stator 3D model; (**b**) zeroth-order mode (7577.4 Hz); (**c**) second-order mode (909.71 Hz); (**d**) third-order mode (2393.4 Hz); (**e**) fourth-order mode (4134.8 Hz); (**f**) fifth-order mode (5682.1 Hz); (**g**) sixth-order mode (6608.8 Hz).

## 5. Electromagnetic Vibration and Noise Analysis

### 5.1. Optimising Scheme Design

This paper used Ansys Maxwell to simulate the 2D model of the motor part of the MEHC, extracted the electromagnetic force into the Harmonic Response module in Workbench for a harmonic response analysis, and imported the results of the harmonic response analysis into the Harmonic Acoustics module for a noise simulation analysis. Two optimisation options for the rotor slotting are proposed. Rotor slotting can change the air-gap magnetic field distribution and be used to reduce the radial electromagnetic force's harmonics, thus reducing motor vibration and noise, as shown in Figure 6.

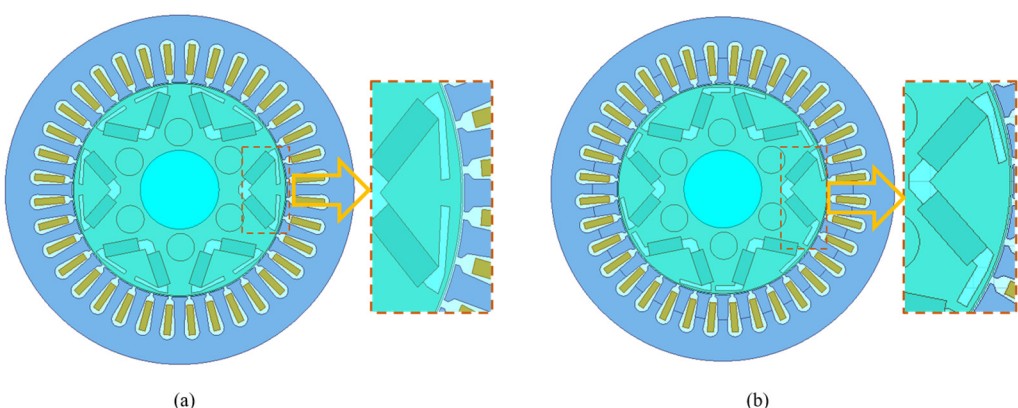

**Figure 6.** Two optimised schemes for rotor slotting: (**a**) scheme one; (**b**) scheme two.

The finite element simulation of the two 2D motor models of the two newly proposed optimised rotor structure solutions and the FFT decomposition of the radial electromagnetic force as a space function is shown in Figure 7a. The FFT decomposition of the radial electromagnetic force amplitude as a function of time after optimisation is shown in Figure 7b, compared with the preoptimisation results.

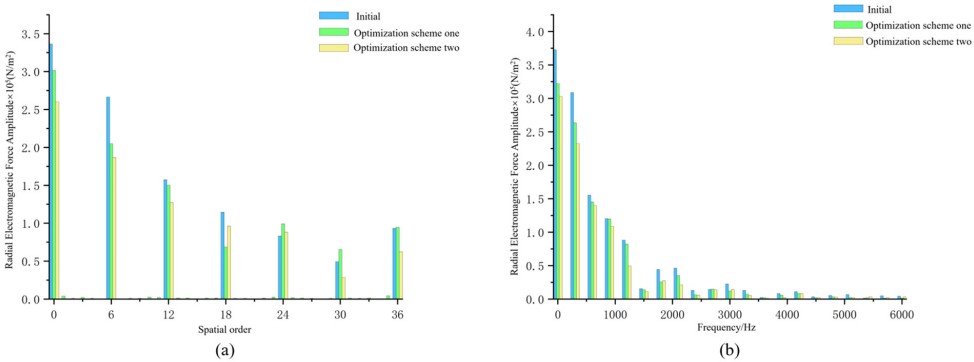

(a) (b)

**Figure 7.** (**a**) Radial electromagnetic force distribution with spatial order; (**b**) radial electromagnetic force distribution with time–frequency at a point on the stator tooth end.

From the comparison results in Figure 7a,b, the spatial and temporal frequency distribution of the optimised radial electromagnetic force is the same as when it is not optimised, except that the amplitude is lower than when it is not optimised, with scheme two generally having the lowest amplitude and being slightly higher at specific harmonic orders. The simulation results illustrate that slotting the rotor can effectively reduce the harmonic amplitude of the radial electromagnetic force, thus reducing motor vibration and noise and improving the NVH performance of the MEHC.

### 5.2. Vibration and Noise Analyses

Because the electromagnetic vibration displacement of a motor is inversely proportional to the fourth power of the spatial order, the effect of higher-order electromagnetic forces on motor vibration and noise can be ignored, and this paper focused on the impact of zeroth-order electromagnetic forces on vibration. The zero-fold fundamental frequency electromagnetic force of the zeroth-order electromagnetic force was removed as it did not cause vibration and noise. A comparison of the zeroth-order EMF spectrum of the motor before optimisation with the two optimisation scenarios is shown in Figure 8.

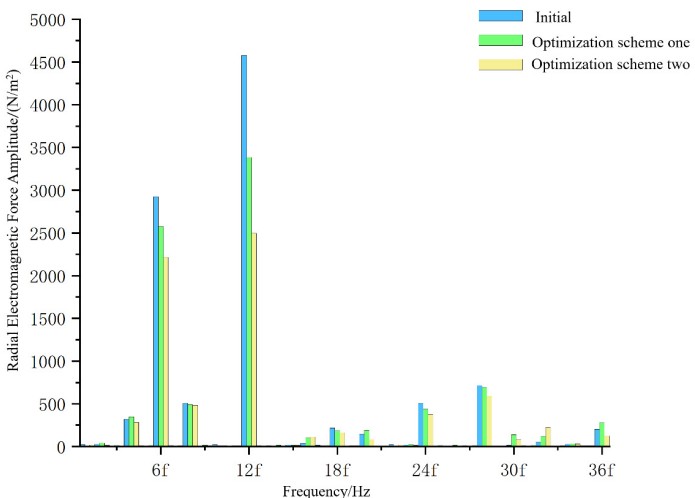

**Figure 8.** Zeroth-order electromagnetic force after removal of zero frequency multiplication.

The analysis of Figure 8 shows that the zeroth-order electromagnetic forces before optimisation and for both optimisation scenarios are mainly larger at 6×the fundamental frequency and 12× the fundamental frequency, so we focused on the 6× fundamental frequency and 12× fundamental frequency for the subsequent vibration and noise analysis.

The radial electromagnetic force was introduced into the three-dimensional model of the stator, and the electromagnetic and structural fields were coupled for the harmonic response analysis. The coupling results are shown in Figure 9, and the vibration acceleration at a point on the stator surface is shown in Figure 10a. The main frequency point amplitudes are shown in Table 4.

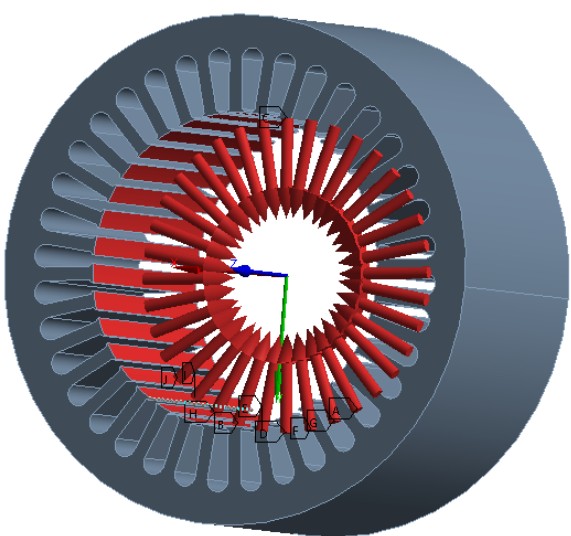

**Figure 9.** Electromagnetic and structural field coupling.

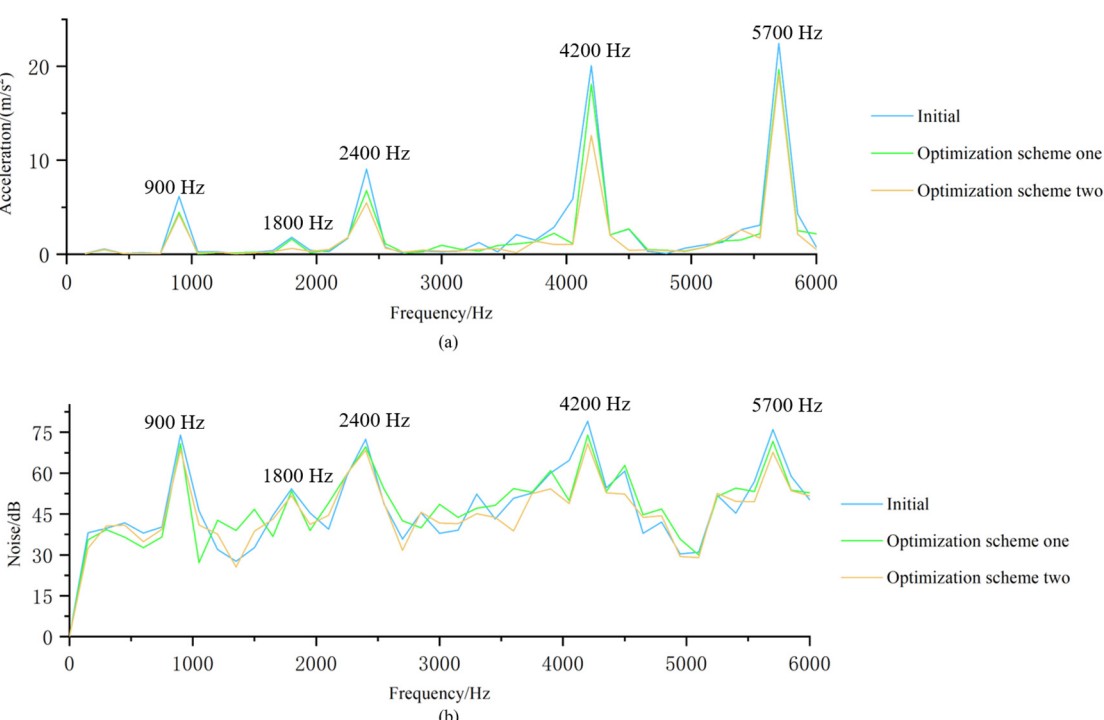

**Figure 10.** (**a**) Vibration acceleration at a point on the stator surface; (**b**) noise at a point in the air domain.

**Table 4.** Vibration acceleration amplitude at major frequency points.

| Frequency/Hz | Initial/(m/s$^2$) | Scheme One/(m/s$^2$) | Scheme Two/(m/s$^2$) |
|---|---|---|---|
| 900 | 6.138 | 4.44 | 4.207 |
| 1800 | 1.811 | 1.601 | 0.6103 |
| 2400 | 9.0547 | 6.7623 | 5.4618 |
| 4200 | 20.056 | 18.084 | 12.64 |
| 5700 | 22.417 | 19.672 | 19.114 |

The analysis of Figure 10a and Table 4 shows that: (1) The vibration acceleration of the motor stator is relatively large at 900 Hz, 2400 Hz, 4200 Hz, and 5700 Hz. At the same time, the amplitude of the vibration acceleration is not too large at 1800 Hz where the electromagnetic force is relatively large. This is because at 900 Hz, it is close to the stator's second-order mode inherent frequency, which causes a forced stator vibration, at 2400 Hz it is close to the stator's third-order mode inherent frequency, which causes a forced stator vibration, and at 4200 Hz it is close to the stator's fourth-order mode inherent frequency, which causes a forced stator vibration. (2) Both optimisation solutions can effectively reduce the vibration acceleration on the stator surface. Scheme 2 is more effective, with the maximum vibration acceleration at 5700 Hz decreasing from 22.417 m/s$^2$ before optimisation to 19.114 m/s$^2$, reducing approximately 14.7%. This is because the slotted rotor affects the magnetic field distribution and weakens the radial electromagnetic force, reducing the vibration acceleration.

The results of the harmonic response analysis were imported into the Harmonic Acoustics module for a noise simulation, and the noise sound pressure level was calculated at a point in the air domain. The results are shown in Figure 10b. The amplitude of the sound pressure level at the main frequency points is shown in Table 5.

**Table 5.** Sound pressure level amplitude at major frequency points.

| Frequency/Hz | Initial/(m/s$^2$) | Scheme One/(m/s$^2$) | Scheme Two/(m/s$^2$) |
|---|---|---|---|
| 900 | 6.138 | 4.44 | 4.207 |
| 1800 | 1.811 | 1.601 | 0.6103 |
| 2400 | 9.0547 | 6.7623 | 5.4618 |
| 4200 | 20.056 | 18.084 | 12.64 |
| 5700 | 22.417 | 19.672 | 19.114 |

Analysis of Figure 10b and Table 5 shows that: (1) The noise fluctuations mainly occur at 900 Hz, 1800 Hz, 2400 Hz, 4200 Hz, and 5700 Hz, consistent with the previous vibration acceleration fluctuation points. Among them, the fluctuation at 1800 Hz is small, mainly caused by the radial electromagnetic force noise, and the noise at the four points of 900 Hz, 2400 Hz, 4200 Hz, and 5700 Hz is mainly caused by the zeroth-order electromagnetic force excitation of the stator's inherent frequency causing a forced vibration, thus generating a larger noise. (2) Compared to the preoptimisation period, both optimisation options effectively reduce noise, with scheme two being more effective, with the maximum noise amplitude dropping from 79.119 dB to 70.86 dB, a reduction of approximately 10.4%. This is consistent with the previous analysis's radial electromagnetic force and vibration acceleration results. In future designs of the motor section of the MEHC, the stator modal inherent frequency should be avoided to avoid resonance and thus improve the NVH performance of the MEHC.

## 6. Conclusions

This paper presented a simulation analysis of the electromagnetic vibration and noise of the MEHC and gave two optimised solutions for rotor slotting, with the following conclusions:

(1) A new type of multisource coupled power device was proposed. This paper selected the Maxwell tensor method to introduce the equation for the radial electromagnetic

force and used electromagnetic simulation software to simulate and analyse the magnetic field of the MEHC motor. We analysed the air-gap magnetic density and the radial electromagnetic force by a fast Fourier transform, and the results were consistent with the analytical method.

(2) Based on analysing the natural frequencies of the stator modes, the vibration of the stator was studied by coupling the electromagnetic field with the structural field. The results revealed that near the natural frequencies of each stator mode, the radial electromagnetic force excited each stator mode to cause a forced vibration, and the motor vibration was large.

(3) Further, this paper simulated the noise of the MEHC motor and selected the noise spectrum of a certain point in the air domain to compare and analyse. Simulation results depicted that where the vibration was larger, the noise fluctuation was also larger. In addition, zeroth-order electromagnetic forces at 12 times the fundamental frequency also caused changes in the noise spectrum.

(4) According to the simulation results of the two optimisation schemes, it was verified that the two methods could effectively reduce the radial electromagnetic force and vibration acceleration and then reduce the noise of the MEHC motor. Among them, the effect of Scheme 2 was superior, which makes it more helpful for improving the NVH performance of MEHC.

**Author Contributions:** Conceptualization, T.Z.; Formal analysis, H.Z.; Investigation, Y.C.; Methodology, B.L. and H.Z.; Resources, T.Z.; Software, B.L.; Writing—original draft, B.L.; Writing—review & editing, B.L. and Z.Z. All authors have read and agreed to the published version of the manuscript.

**Funding:** National Natural Science Foundation of China: 52075278.

**Conflicts of Interest:** The authors declare no conflict of interest.

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
