# Peer review of "Simulation Analysis on Electromagnetic Vibration and Noise of Novel Mechatronic-Electro-Hydraulic Coupler"

_machines, doi:10.3390/machines10090762_

Round 1

Reviewer 1 Report

This paper presented two types of rotor structure which can improve noise and vibration characteristics. It is verified through coupling analysis using ANSYS/Maxwell. You should respond to the following comments thoroughly for artcle publication.

1. the two optimised structures have good performances on reduing electromagetic vibration. However, the comparison results shown in Figs. 7 and 8 are "unfair" because the two optimised structure have lower torque constant than the initial structure, due to the slotting (flux bridge). Please comment on this problem.

2. The scheme #2 have assymetric bridge which oriented in one direction. This leads to different motor performance when the motor rotate clockwise and counterclockwise. Please comment.

Reviewer 2 Report

This paper presents a finite element method-based analysis of harmonics in the electromagnetic field distribution of PM-assisted synchronous reluctance motor. The topic is interesting and very significant for electric vehicles. The optimization of electric machines is not a straightforward and easy task. By changing one parameter, several state variables of the machine may change due to the coupling effect.

Please improve the technical detail of your optimization method. No doubt the optimization is difficult on a hit-and-trial basis. 

Could you please include your optimization algorithm?

Moreover, as the results are based on FEM analysis, the analytical model seems for theoretical discussion. If so please try to make it easily understandable for the reader new to this field.  

For example, what are "theta" and "t" variables and what is the Maxwell tensor?

Several other factors can also enhance spatial harmonics such as the switching harmonics from the inverter, unbalanced windings, material saturation etc. How those problems can be addressed?

Round 2

Reviewer 1 Report

Authors have addressed all my comments throughly. I hope your publication.

Reviewer 2 Report

Thanks for answering the questions